# Urea Cycle Related Amino Acids Measured in Dried Bloodspots Enable Long-Term In Vivo Monitoring and Therapeutic Adjustment

**DOI:** 10.3390/metabo9110275

**Published:** 2019-11-12

**Authors:** Julien Baruteau, Youssef Khalil, Stephanie Grunewald, Marta Zancolli, Anupam Chakrapani, Maureen Cleary, James Davison, Emma Footitt, Simon N. Waddington, Paul Gissen, Philippa Mills

**Affiliations:** 1Genetics and Genomic Medicine, Great Ormond Street Institute of Child Health, University College London, London WC1E 6BT, UK; y.khalil@ucl.ac.uk (Y.K.); p.gissen@ucl.ac.uk (P.G.); p.mills@ucl.ac.uk (P.M.); 2Gene Transfer Technology Group, Institute for Women’s Health, University College London, London WC1E 6BT, UK; s.waddington@ucl.ac.uk; 3Metabolic Medicine Department, Great Ormond Street Hospital for Children NHS Foundation Trust, London WC1N 3JH, UK; stephanie.grunewald@gosh.nhs.uk (S.G.); anupam.chakrapani@gosh.nhs.uk (A.C.); maureen.cleary@gosh.nhs.uk (M.C.); james.davison@gosh.nhs.uk (J.D.); emma.footitt@gosh.nhs.uk (E.F.); 4National Institute of Health Research Great Ormond Street Hospital Biomedical Research Centre, London W1T 7HA, UK; marta.zancolli@gosh.nhs.uk; 5Wits/SAMRC Antiviral Gene Therapy Research Unit, Faculty of Health Sciences, University of the Witwatersrand, Johannesburg 2000, South Africa; 6MRC Laboratory for Molecular Cell Biology, University College London, London WC1E 6BT, UK

**Keywords:** tandem mass spectrometry, dried blood spots, urea cycle, amino acids, argininosuccinate lyase

## Abstract

Background: Dried bloodspots are easy to collect and to transport to assess various metabolites, such as amino acids. Dried bloodspots are routinely used for diagnosis and monitoring of some inherited metabolic diseases. Methods: Measurement of amino acids from dried blood spots by liquid chromatography-tandem mass spectrometry. Results: We describe a novel rapid method to measure underivatised urea cycle related amino acids. Application of this method enabled accurate monitoring of these amino acids to assess the efficacy of therapies in argininosuccinate lyase deficient mice and monitoring of these metabolites in patients with urea cycle defects. Conclusion: Measuring urea cycle related amino acids in urea cycle defects from dried blood spots is a reliable tool in animal research and will be of benefit in the clinic, facilitating optimisation of protein-restricted diet and preventing amino acid deprivation.

## 1. Introduction

Liquid chromatography-tandem mass spectrometry (LC-MS/MS) is a technique routinely used in clinical laboratories enabling inexpensive high-throughput identification and accurate quantification of metabolites in limited volumes of biological fluids. Dried blood spots (DBS) are a simple method of blood collection and are easily transportable by post. Quantification of compounds from DBS with LC-MSMS has been supported worldwide with a broad field of applications, particularly for new-born screening [1], diagnosis [2,3], and monitoring of inherited metabolic diseases (IMD), such as phenylketonuria [4], homocystinuria [5], and maple syrup urine disease [6].

Amino acids are used as surrogate biomarkers for various IMD. In urea cycle defects, analysis of intermediary amino acids of the urea cycle, i.e., ornithine, citrulline, argininosuccinic acid, and arginine, are critical in aiding the diagnosis, optimising the protein-restricted diet, and amino acid supplementation of patients. High glutamine levels may indicate hyperammonaemia [7] and are predictive of hyperammonaemic crises [8]. All of these amino acids have been investigated as biomarkers for new-born screening in urea cycle defects [9,10].

Methods to measure amino acids have progressed from time-consuming ion-exchange chromatography with post-column ninhydrin derivatisation [11]. High performance liquid chromatography [12] and tandem mass spectrometry (MS/MS) [10] usually require with precolumn derivatisation, e.g., with butyl esterification, requiring manual processing hence another time consuming step, which can potentially affect the reproducibility of the technique. Coupling LC and MS/MS enables a good separation, enabling accurate detection of underivatised amino acids, although the ion-pairing reagent might cause ion suppression [13,14,15].

In this work, we present a LC-MS/MS method optimised to measure underivatised urea cycle related amino acids, its application in vivo to assess efficacy of therapies in argininosuccinate lyase deficient *Asl^Neo/Neo^* mice, and to monitor urea cycle deficient patients. 

## 2. Results 

### 2.1. Dried Blood Spot Extraction

Optimisation of amino acid extraction from dried blood spot was investigated. The volume of eluent (i.e., methanol) was progressively increased from 100 μL to 500 μL. This did not affect the concentration of analytes detected (Appendix A). The time of extraction was increased from 15 min to 7 h. A prolonged extraction time did not show any obvious benefit for the amino acids tested with the exception of L-ornithine, which showed a 3-fold increase in elution efficiency with longer time periods (*p* < 0.01) (Appendix A). The use of 3.2 mm and 6.4 mm diameter punches, corresponding to a blood volume of 3.4 μL and 11.2 μL, respectively [1], did not show any significant variation in the extraction rate for these amino acids (Appendix A). Sonication of the DBS punches did not affect extraction efficiency (Appendix A).

### 2.2. Method Validation 

Multiple reaction monitoring using cone voltage and collision energies, which resulted in maximal ion detection for both precursor and product ions, were identified (Appendix A). Linearity was observed for all amino acids between 0 to 200 μM (Figure 1). Precision studies (Table 1) showed intra- and inter-batch coefficients of variation (CVs) all below 10% except for argininosuccinic acid. The intra-batch CV of argininosuccinic acid was below 10%, whilst the inter-batch CV was below 15% at medium and high concentrations (300 and 1500 μM, respectively). At low concentrations (50 µM) however, the value was 27%. The limit of detection (LOD) and limit of quantification (LOQ) were calculated using the standard deviation method and measured between 0.2 to 2.3 μM and 0.8 to 7.7 μM, respectively, for all amino acids. Matrix effects were negligible for l-arginine (85%), l-citrulline (98%), l-ornithine (104%), and l-glutamate (93%), but showed potentiation and ion suppression for l-glutamine (140%) and argininosuccinic acid (33%), respectively (Appendix A). Recovery experiments indicated that the internal standard dilution method was able to compensate for these matrix effects [16].

### 2.3. Monitoring of Urea Cycle Related Amino Acids in Asl^Neo/Neo^ Mice

We assessed levels of arginine, citrulline, argininosuccinic acid, and its anhydrides alongside the combined concentrations of glutamate and glutamine in argininosuccinate lyase (ASL) deficient *Asl^Neo/Neo^* and wild-type mice. This ASL-deficient mouse model recapitulates the human phenotype of inherited ASL deficiency also referred to as argininosuccinic aciduria. Levels were assessed at three time points, i.e., 2-, 3-, and 4-month-old animals and showed reduced arginine, increased citrulline, argininosuccinic acid, and glutamine/glutamate levels with limited variation overtime (Figure 2).

### 2.4. Monitoring of Glutamine and Glutamate Levels in Dried Blood Spots as A Surrogate of Ammonia Levels in Asl^Neo/Neo^ Mice

Glutamate and glutamine buffer ammonia and can therefore be used as an indirect biomarker of ammonia levels. We measured cumulated levels of glutamate and glutamine, and correlated this with plasma ammonia levels in WT and untreated *Asl^Neo/Neo^* mice. The glutamate-glutamine levels correlated with ammonaemia levels (r_s_ = 0.74; *p* < 0.0001) (Figure 3).

### 2.5. Monitoring of Arginine Supplementation in Asl^Neo/Neo^ Mice

Arginine levels were measured in dried blood spots from WT, untreated, and arginine supplemented *Asl^Neo/Neo^* mice. We observed a significant decrease in the concentration of the arginine levels in untreated *Asl^Neo/Neo^* mice compared to that of WT (*p* < 0.001). This decrease improved with arginine supplementation *(p* < 0.05) (Figure 4).

### 2.6. Urea Cycle Related Amino Acid Levels in Dried Blood Spots from Patients with Urea Cycle Defects 

Levels of arginine, citrulline, argininosuccinic acid and its anhydrides, and summed glutamate and glutamine (Figure 5) were assessed in dried blood spots collected from urea cycle deficient patients. These patients were metabolically stable and included individuals with ornithine transcarbamylase deficiency (OTCD) (*n* = 3), argininosuccinate synthase deficiency (ASSD) (*n* = 3), argininosuccinate lyase deficiency (ASLD) (*n* = 2), and N-acetylglutamate synthase deficiency (NAGSD) (*n* = 1).

As anticipated and compared to healthy controls, citrulline levels were dramatically elevated in ASSD, slightly increased in ASLD patients, and in a similar range or slightly decreased in OTCD. The pathognomonic biomarker argininosuccinic acid was only found in ASLD patients. Cumulated glutamate and glutamine levels were similar between disorders and were all below <1000 μM except for one ASLD patient with concomitant borderline ammonia level (57 μM).

## 3. Discussion

This report describes the development and in vivo application of a rapid LC-MS/MS method to accurately monitor six urea cycle related amino acids from dried blood spots, i.e., ornithine, citrulline, argininosuccinic acid, arginine, glutamine and glutamate.

Optimisation of methods for the extraction of amino acids from dried blood spots is an essential, but challenging step as incomplete elution or analyte degradation can cause inaccurate results. Amino acid extraction from DBS has been reported previously and is commonly performed as an off-line process [17] at room temperature with methanol, a generic solvent [10]. Temperature and extraction solution (including pH) were therefore not investigated during this study. However, published methods of amino acid extraction from DBS have no consensus regarding the parameters investigated in this work. Extraction times vary from 20–60 min [18,19,20], volumes of methanol used range from 100–900 μL [19], and punch sizes vary from 3.2–6.4 mm (1/4 inches) in diameter [20]. Sonication is incorporated into some protocols [15,21,22]. In our work, only marginal variations were observed by modifying extraction parameters, i.e., time, volume of eluent, punch size, additional solvation energy (sonication) for all of the amino acid analysed with the exception of ornithine, which showed a 2- to 3-fold increase in elution efficiency after 2 and 7 h of extraction, respectively.

This method adapted from Piraud et al. [23] does not involve derivatisation, an approach, which has been reported to be more precise for the analysis of amino acids of the urea cycle [24]. Another advantage of non-derivatisation is the decreased time involved in sample preparation. The liquid chromatography gradient used for the LC-MSMS amino acid analysis incorporates a high percentage of acetic acid in its mobile phase, similar to that used previously for the analysis of B_6_ vitamers [25]. The method shows good linearity over the physiological range of analytes. The precision and reproducibility of the method, including extraction and analysis, shows good intra-batch reproducibility for all compounds and a satisfactory inter-batch reproducibility for all of the compounds analysed with the exception of argininosuccinic aciduria when present at low concentrations (50 μM). However, this remained within acceptable values published for dried blood spot analysis [24,26] and within the range of argininosuccinic acid levels usually observed in untreated Asl^Neo/Neo^ mice [27] and ASLD patients [28].

Compared to others, our method is quick, accurate, and cost-effective with the advantage of using dried blood spots, i.e., requiring minimal sample as low as 20 μL The overall extraction-detection process of our method remains relatively fast, i.e., 15 min and 25 min for extraction and LC-MS/MS detection, respectively. LC-MS/MS methods usually used for new-born screening are now using either butyl ester derivatisation [29] or underivatised [15] approaches. The development of faster methods with more sensitive mass spectrometers and reduced steps has enabled a quicker turn round to maximise efficacy. Liquid chromatography for separation is not always required. The running time in that case can be as quick as 1 min [15], enabling high throughput processing at minimal cost.

Amino acid analysis of dried blood spots has been used to routinely monitor and optimise therapy for the treatment of certain inherited metabolic diseases without (e.g., phenylketonuria, homocystinuria [4,5]) and with (e.g., MSUD [6]) acute decompensations. This study suggests that this approach could be considered for urea cycle defects. Applications could include research in small animal models [27], clinical monitoring to adjust the diet or amino acid supplementation, and identification of patients with higher risk of hyperammonaemic decompensation.

It is known that ureagenesis, ammonia, and amino acids levels are influenced by fed or fasted states. Monitoring in outpatient clinics does not always allow blood sampling at comparable times, which can make the interpretation of results more difficult. Sampling at a similar time of the day performed by the patient could allow a more reliable comparison. This would be of interest as well for other amino acids, such as the branched chain amino acids, which have been reported to be low in some urea cycle deficient patients [30]. Implementation of a more reliable regular monitoring could help clinicians and specialised metabolic dieticians to tailor the protein-restricted diet and amino acid supplementation given to urea cycle patients. For example, regular monitoring of arginine could help preventing secondary creatine deficiency [31]. Monitoring of leucine and valine, which have been recently associated with faltering growth in urea cycle defects [32], could help to optimise supplementation to achieve physiological levels. Careful monitoring of glutamate and glutamine could identify patients with higher risk of decompensations [33].

In some specific cases in developing countries with limited resources and where patients live in remote locations, this approach could help aid diagnosis and guide genetic investigations. However, it is essential to emphasize that whilst this approach can complement the monitoring of therapies for treatment of urea cycle defects it cannot be used as a surrogate for ammonia level monitoring in an acute clinical situation with a patient at risk of hyperammonaemia.

## 4. Materials and Methods

### 4.1. Materials

l-Arginine-13C and l-Glutamine-1,2-13C2 from CK isotopes. Ibstock, UK, l-Citrulline-2,3,3,4,4,5,5-d7, l-Glutamic-2,3,3,4,4-d5 Acid, and l-Ornithine-2,3,3,4,4,5,5-d7 HCl from CDN isotopes. Pointe-Claire, Quebec, and Canada were used as internal standards. l-Arginine, l-Citrulline, l-Glutamine, l-Glutamic Acid, l-Argininosucciniate, and l-Ornithine were obtained from Sigma, St Louis, MO, USA.

### 4.2. Blood Spot Collection and Sample Preparation

Extraction of amino acids from DBS of anonymised healthy adult controls was adapted from Schulze et al. [34]. Whole blood was spiked onto a Guthrie card at the time of collection and allowed to dry at room temperature for 24 h before being stored at −20 °C in a foil bag with desiccant. On the day of analysis, a punch from the centre of the dried blood spot was placed in a 1.5 mL Eppendorf, which contained methanol and a mixture of internal standards (2 nM of l-Arginine-13C, l-Citrulline-2,3,3,4,4,5,5-d7, and l-Glutamic-2,3,3,4,4-d5 Acid, and 0.66 nM l-Glutamine-1,2-13C2 and l-Ornithine-2,3,3,4,4,5,5-d7). The supernatant was removed and centrifuged at 16,000× *g* for 5 min prior to injection of 3 µL into the LC-MSMS system.

### 4.3. LC-MSMS

The stationary phase used was a reversed-phase hybrid-based C18 column (Waters XTerra^®^ RP_18_). A 5 μm 3.9 × 150 mm column was used (Waters, Milford, MA, USA), fitted with the appropriate guard column. A Waters Alliance 2795 LC system (Waters, Midford, MA, USA) high-performance liquid chromatography machine (HPLC) was used to separate the analytes of interest with a 25 min gradient profile modified from Footitt et al. [25]. Two mobile phases, (A) methanol and (B) 3.7% acetic acid, were used with 0.2mL/min flow rate according to the following gradient profile: Phase A: 85% for minute 1–6, 75% for minute 6–8; 5% for minute 9–15, 100% for minute 16–25; 100% B for the first minute, then gradual increase. The column was reconditioned for 10 min at the end of each run.

Detection used a Micromass Quattro micro API (Micromass UK Ltd., Cheshire, UK) tandem mass spectrometer using multiple reaction monitoring in positive ion mode. The temperature of the source and for desolvation were 120 °C and 350 °C, respectively. The capillary and cone voltages are described in Appendix A. The cone gas flow was 50 L/h. The syringe pump flow was set at 30 µL/min. The mass spectrometer vacuum was 4.3 × 10^−3^ mbar. The multiplier and extractor voltages were 650 Volts and 1 Volt, respectively. Dry and highly purified argon was used as the collision cell gas. Data were analysed using Masslynx 4.1 software (Micromass UK Ltd., Cheshire, UK). For quantification, standard curves of stable isotopes were compared to internal standards spiked in samples.

### 4.4. Analysis

Glutamine can be converted into glutamate either spontaneously or via the catalytic activity of glutaminase [35]. This spontaneous deamidation can occur during storage [36]. Taking this natural deamination into account, it was decided to combine the results for glutamate and glutamine. Similarly, the results for argininosuccinic acid and its anhydrides were combined. Argininosuccinic acid is capable of creating a structure with a five-membered ring when an acyl group binds an amine group located within the aliphatic chain of the molecule. This results in the loss of one molecule of water, creating an anhydride derivative.

Precision studies were performed according to international guidelines (International Conference on Harmonisation, Harmonised Tripartite Guideline, and Validation of Analytical Procedures: Text and Methodology Q2(R1). 1994). Whole blood was spiked onto a Guthrie card with 3 different known concentrations for each amino acid of interest: low (50 μM), medium (300 μM), and high (1500 μM). Reproducibility was assessed by inter- and intra-assay CV. The inter-assay CV was established by performing 5 assays spaced by >7 days apart. The intra-assay CV was established by 10 repeats on the same day. Recovery was calculated on 10 samples run on the same day.

### 4.5. Animals

The argininosuccinate lyase (ASL) deficient *Asl^Neo/Neo^* mice (B6.129S7-Asltm1Brle/J) were purchased from Jackson Laboratory (Bar Harbor, ME, USA). Wild-type [37] and *Asl^Neo/Neo^* littermates were maintained on standard rodent chow (Harlan 2018, Teklab Diets, Madison, WI, USA; protein content 18%) with free access to water. WT and *Asl^Neo/Neo^* littermates were housed in the same cages. Mouse experiments were approved by institutional ethical review and performed under UK Home Office licences 70/6906 and 70/8030.

### 4.6. Ethics

Sampling of blood from patients with urea cycle defects was approved by the National Research Ethics Service Committee London-Bloomsbury (13/LO/0168).

### 4.7. Statistics

Data were analysed using GraphPad Prism 5.0 software, San Diego, CA, USA. The figure shows mean ± standard deviation (SD). Comparisons of continuous variables between three or more experimental groups were performed using the One-way ANOVA test with Dunnett’s post-test for pairwise comparison. *p* values <0.05 were considered statistically significant.

## 5. Conclusions

This work presents a rapid and sensitive method to measure underivatised urea cycle related amino acids in tiny samples as dried blood spots. The described method was optimised to allow the best extraction and detection without the additional step of derivatisation. Optimal separation was obtained with LC column which makes this method slightly slower compared to more recently described methods. Examples of in vivo application show the potential of this assay both for clinical and research purposes as a way to provide better monitoring of the protein restricted diet of Urea Cycle patients and for assessing efficacy of therapies.

## Figures and Tables

**Figure 1 metabolites-09-00275-f001:**
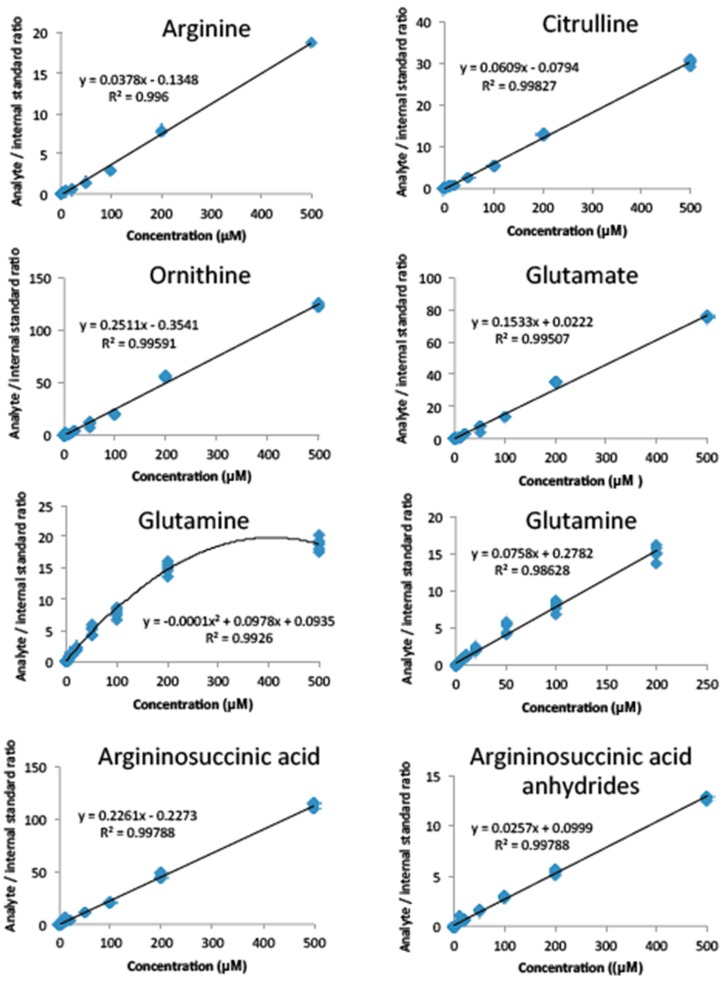
Linearity of measured amino acids in dried blood spots. Five calibration curves in methanol are presented. Arginine, citrulline, glutamate, ornithine, and argininosuccinic acid and argininosuccinic acid anhydrides were shown to be linear between 0 to 500 μM and glutamine from 0 to 200 μM.

**Figure 2 metabolites-09-00275-f002:**
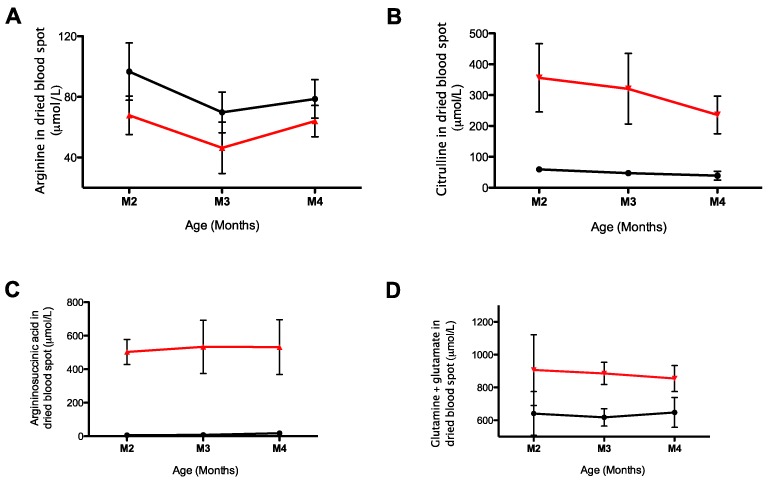
Monitoring of urea cycle-related amino acids overtime in *Asl^Neo/Neo^* mice. (**A**) Arginine, (**B**) citrulline, (**C**) argininosuccinic acid and anhydrides, and (**D**) glutamine and glutamate in WT (*n* = 8–18), and Asl^Neo/Neo^ (*n* = 5–10) littermates. Graphs represent mean ± SD.

**Figure 3 metabolites-09-00275-f003:**
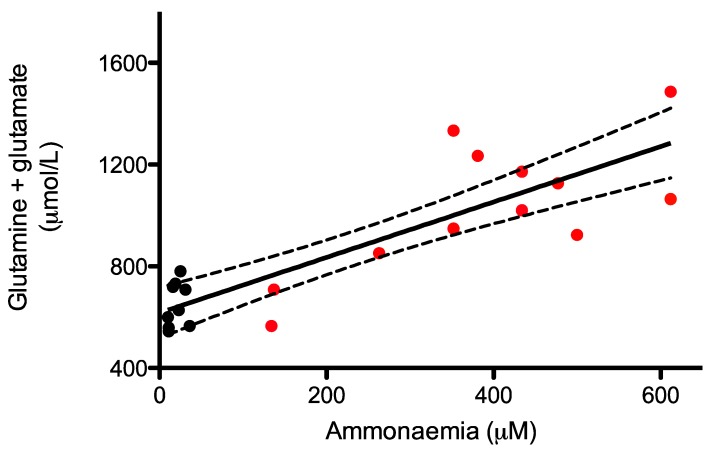
Correlation between ammonaemia versus glutamine and glutamate levels in dried blood spots. Parameters were measured in wild type (black; *n* = 9) and untreated Asl^Neo/Neo^ (red; *n* = 15) littermates. Spearman’s rank correlation coefficient test (r_s_ = 0.74; *p* < 0.0001).

**Figure 4 metabolites-09-00275-f004:**
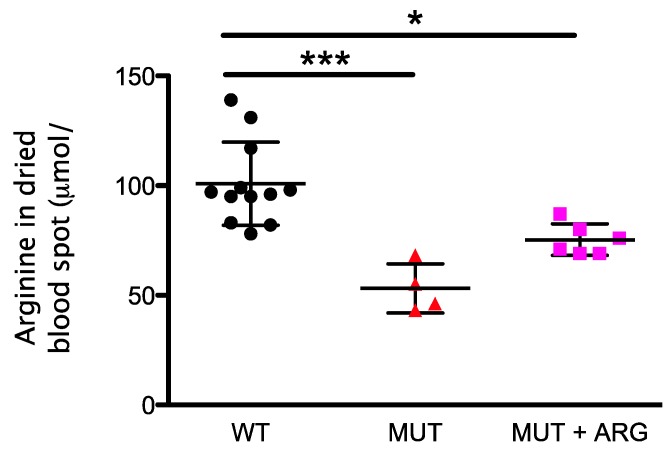
Monitoring of arginine levels in dried blood spot. Samples were collected from wild type (WT, *n* = 12), argininosuccinate lyase deficient littermates untreated (MUT, *n* = 4), or treated with daily intraperitoneal injections of L-arginine (1 g/kg/d) (MUT + ARG, *n* = 6). Samples were collected at 3–8 weeks of life. Graphs represent mean ± SD.

**Figure 5 metabolites-09-00275-f005:**
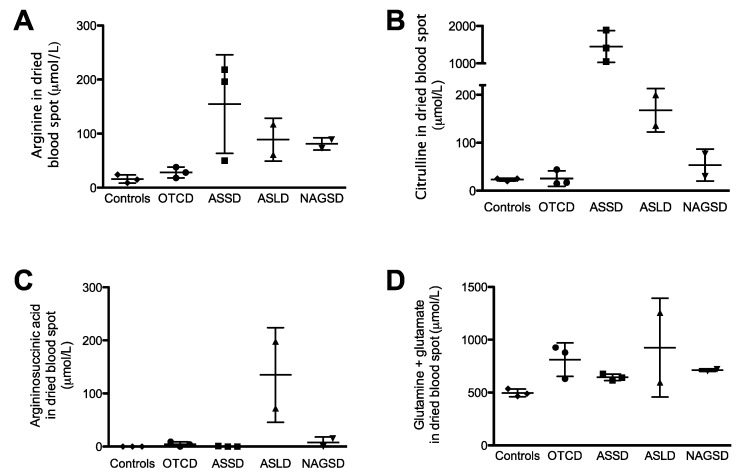
Monitoring of urea cycle-related amino acids in patients with urea cycle defects. (**A**) Arginine, (**B**) citrulline, (**C**) argininosuccinic acid and anhydrides, (**D**) summed glutamine and glutamate in ornithine transcarbamylase deficiency (OTCD, *n* = 3), argininosuccinate synthase deficiency (ASSD, *n* = 3), argininosuccinate lyase deficiency (ASLD, *n* = 2), and N-acetylglutamine synthase deficiency (NAGSD, *n* = 1 measured at 2 time points). Graphs represent mean ± SD.

**Table 1 metabolites-09-00275-t001:** Recovery, intra- and inter-batch coefficient of variations for each analyte.

Amino acid Amound Added to Whole Blood	50 μM			300 μM			1500 μM		
Parameter tested	Recovery (%)	Intrabatch CV (%)	Interbatch CV (%)	Recovery (%)	Intrabatch CV (%)	Interbatch CV (%)	Recovery (%)	Intrabatch CV (%)	Interbatch CV (%)
**Arginine**	99	5	5	98	2	4	97	6	5
**Citrulline**	104	2	9	116	3	2	106	7	3
**Ornithine**	99	5	3	101	3	2	99	3	3
**Glutamate**	101	2	6	107	2	2	102	4	4
**Glutamine**	99	1	0	99	2	2	96	2	1
**Argininosuccinic acid and anhydrides**	74	9	27	110	3	14	103	0	12

Three different concentrations were measured: low 50 μM, medium 300 μM, and high 1500 μM. Amino acids were spiked onto a Guthrie card. CV: coefficient of variance.

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
