# Peer review of "Urea Cycle Related Amino Acids Measured in Dried Bloodspots Enable Long-Term In Vivo Monitoring and Therapeutic Adjustment"

_metabolites, 2019, doi:10.3390/metabo9110275_

Round 1

Reviewer 1 Report

The authors describe a method to measure a few amino acids, concerning with urea cycle disorders, in dried blood spot (DBS). The method described is in detail. They also tested their method by mouse and patient samples. They did a good correlation between DBS glutamine+glutamic to blood ammonia. However, there are a few general criticism to the manuscript.

DBS amino acid (and acylcarnitines) measurement is a routine in newborn screening, and the commonly used kit also employ non-derivative method. The flow injection method is faster than the method in this article. The authors compared a few DBS elution methods, but the LC program is slow. There has been papers describing LSMSMS measurements for amino acids. The authors may want to compare their method with previous publications. Alteration of amino acid in urea cycle disorder is quite well known. Therefore, it is difficult to understand the value in the mouse and human samples.

Author Response

We thank the reviewer for his comment. 

We have added a paragraph in the discussion section comparing our method to other publications. We have as well added more information in the introduction about methods usually used to measure amino acids to better highlight the interest of our method. 

The modifications of amino acids in urea cycle defects reported in our manuscript are not supposed to be novel findings but are usually measured in plasma. This manuscript shows that these modifications can be effectively monitored in dried blood spots from mouse and human samples with a range of potential applications proposed in the discussion.

Reviewer 2 Report

Review of manuscript ID: metabolites-616764 “Urea cycle related amino acids measured in dried bloodspots enable long-term in vivo monitoring and therapeutic adjustment.”

SUMMARY
In this manuscript, the authors present a novel method for the accurate and rapid determination of several amino acids in dried bloodspots samples

COMMENTS

In my opinion, this manuscript describes correctly the methodology used by the authors and the results obtained seem sound. However, I think that some issues should be corrected before its publication.

First, I miss a more in-depth introduction describing the state-of-the-art of the DBS analysis. Which other methods are used? What are the figures of merit of these methods? Regarding LC-MS/MS, what are the experimental conditions of the most used methods? All these new details will allow understanding of the main advantages of the proposed method better. For instance, part of the discussion can be moved to the introduction.

Results

Section 2.1. I feel that the design of this study does not allow to find the optimal conditions for the DBS extraction. The assessment of one variable at a time does not allow to evaluate possible interaction between the considered factors (volume, time, punch size, sonication, …). Are the authors sure that all these variables are totally independent? If not, the experimental design should be employed.

Figure 1. It is difficult to evaluate the linear behaviour of the different compounds. Please, consider changing the graph to a more informative plot and adding other figures of merit.

Section 2.5 and 4.7. Please, give more details regarding the statistical method used and if the different assumptions for the application of the two-tailed t-test are fulfilled.

Methods

Line 191. Which other compounds? Please, specify.
Conclusion

Please, rewrite the paragraph highlighting the main advantages of the method and the potential drawbacks.

Author Response

COMMENTS

In my opinion, this manuscript describes correctly the methodology used by the authors and the results obtained seem sound. However, I think that some issues should be corrected before its publication.

First, I miss a more in-depth introduction describing the state-of-the-art of the DBS analysis. Which other methods are used? What are the figures of merit of these methods? Regarding LC-MS/MS, what are the experimental conditions of the most used methods? All these new details will allow understanding of the main advantages of the proposed method better. For instance, part of the discussion can be moved to the introduction.

We thank the reviewer for his comment. The introduction has been amended presenting the current methods and their advantages and respective drawbacks to better highlight the advantage of our method.  

Results

Section 2.1. I feel that the design of this study does not allow to find the optimal conditions for the DBS extraction. The assessment of one variable at a time does not allow to evaluate possible interaction between the considered factors (volume, time, punch size, sonication, …). Are the authors sure that all these variables are totally independent? If not, the experimental design should be employed.

There are a limited number of publications detailing optimisation of the extraction from dried blood spots. These publications focus on the different composition and pH of the elution buffer (for example Wang et al, Amino Acids 2013:reference 15; Han et al, BioRxiv doi10.1101/195295) but do not assess volume, time or sonication effect. When this is done, this is performed sequentially as described in our manuscript (Wuyts et al Clin Chim Acta 2007; reference 24).  

Figure 1. It is difficult to evaluate the linear behaviour of the different compounds. Please, consider changing the graph to a more informative plot and adding other figures of merit.

We thank the reviewer for his comment. The graphs have not been merged together but aree now presented individually. For glutamine, 2 graphs are shown. A first one shows the saturation effect >200uM; a second one highlights the linearity observed from 0-200uM.

Section 2.5 and 4.7. Please, give more details regarding the statistical method used and if the different assumptions for the application of the two-tailed t-test are fulfilled.

We thank the reviewer for his comment. We recognise that the comparison of 3 groups will require a One way ANOVA test rather than a Student’s t test. Statistical calculations have been changed accordingly and the method section has been amended. 

Methods

Line 191. Which other compounds? Please, specify.

This has been added.

Conclusion

Please, rewrite the paragraph highlighting the main advantages of the method and the potential drawbacks.

Thank you for your comments. This has been corrected.

Round 2

Reviewer 2 Report

The manuscript metabolites-616764 "Urea cycle related amino acids measured in dried bloodspots enable long-term in vivo monitoring and therapeutic adjustment." by Baruteau et al. has been sufficiently improved to be suitable for publication.

The authors have addressed most of the concerns I had with the previous submission. I am still not convinced of the approach used for the parameter optimization regardless of whether it is used in other published papers. The potential interaction between factors should always be considered, and the “one-at-a-time” strategy does not allow it.